# Quantum electrometer for time-resolved material science at the atomic lattice scale

Gregor Pieplow[1,3], Cem Güney Torun ®[1,3], Charlotta Gurr[1], Joseph H. D. Munns[1], Franziska Marie Herrmann ®[1], Andreas Thies[2], Tommaso Pregnolato ®[1,2] & Tim Schröder ®[1,2] ✉

The detection of individual charges plays a crucial role in fundamental material science and the advancement of classical and quantum high-performance technologies that operate with low noise. However, resolving charges at the lattice scale in a time-resolved manner has not been achieved so far. Here, we present the development of an electrometer with 60 ns acquisition steps, leveraging on the spectroscopy of an optically-active spin defect embedded in a solid-state material with a non-linear Stark response. By applying our approach to diamond, a widely used platform for quantum technology applications, we can distinguish the distinct charge traps at the lattice scale, quantify their impact on transport dynamics and noise generation, analyze relevant material properties, and develop strategies for material optimization.

Free charge carriers such as electrons are essential components of our modern-day world. They facilitate devices such as smartphones and computers. Uncontrolled or undesired charges, on the other hand, can cause damage and reduce the performance of such devices. Prominent examples are gate-oxide breakdown in flash memory[1] and charge-noise on the nanoscopic level[2,3]. The detection and quantification of desired and undesired charge carriers with electrometers[4,5] holds significant technological importance on the nanoscale.

Despite significant progress[5–16], electrometers have to date been unable to provide time-resolved access to elementary charges with subnanometer resolution. Precisely resolving charges and performing temporal analysis at atomic lattice scales, however, is becoming increasingly important. For example, the investigation of 2D ferro-electric systems would greatly benefit from the use of a highly sensitive electrometer, which could provide critical insights into the unresolved foundational aspects of their physical characteristics[17]. Furthermore, as silicon transistors reduce to sizes of a few nanometers they become more susceptible to charge-induced noise[18,19].

Particularly, quantum technology applications face challenges: in ion-based quantum computers, localized electronic states are suspected to cause decoherence due to motional heating[20]; super-conducting qubits suffer from defect-induced charge-noise[21,22]; in atom-like spin qubits in wide-bandgap semiconductors, charge-noise

leads to optical and spin decoherence[5,13,23–25], significantly limiting the development of quantum networking and sensing[26,27].

Understanding the underlying mechanisms of such platform-specific detrimental processes is a necessity to improve the performance and application range of nanoscale electronic and photonic devices including open questions around decoherence processes, electron dynamics, and material questions of lattice defect formation.

Here, we introduce a quantum electrometer that enables the detection of electric fields produced by single and multiple elementary charges with a relative sensitivity of $10^{-7}$ and resolves individual charge state dynamics down to tens of nanoseconds at the Ångström-scale.

The electrometer consists of an electric field-sensitive, optically active, local probe and a read-out unit (Fig. 1a). In this demonstration, the local probe is a negatively charged tin-vacancy color center (SnV) in diamond[28], a solid-state defect with fluorescent transitions and a non-linear electric field response, which is typical for defects in the $D_{3d}$ point group[28–33]. The optical transition energies directly depend on the local electric field via the DC-Stark-effect $\Delta_{\text{Stark}} = -\mu_{\text{ind}}(E_s)E_s$, where $\mu_{\text{ind}}$ is the induced dipole moment of the atomic defect and $E_s$ is the sum of all static electric fields produced by surrounding charges, which shift the optical energies[34] (Fig. 1b).

The read-out unit is a microscope which is used to perform pho-toluminescence excitation spectroscopy on the local atomic probe,

[1]Department of Physics, Humboldt-Universität zu Berlin, 12489 Berlin, Germany. [2]Ferdinand-Braun-Institut (FBH), Gustav-Kirchhoff-Str. 4, 12489 Berlin, Germany. [3]These authors contributed equally: Gregor Pieplow, Cem Güney Torun. ✉e-mail: tim.schroeder@physik.hu-berlin.de

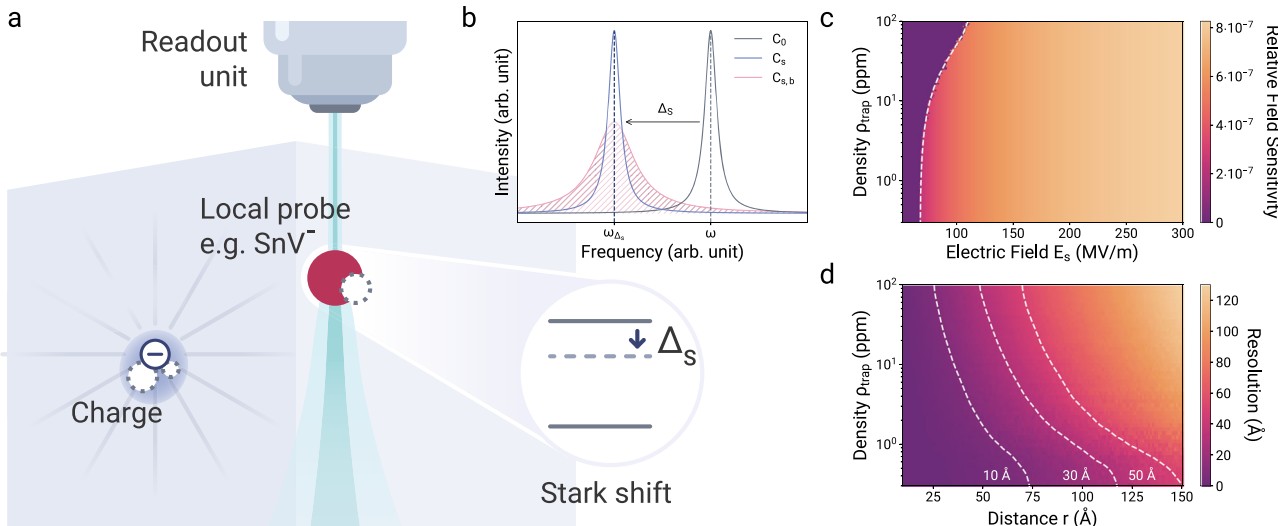

**Fig. 1 | Working principle of the quantum electrometer. a** The local probe is an optically active atomic defect with non-linear Stark-sensitive energy levels. The read-out unit is a photoluminescence excitation spectroscopy microscope. **b** A nearby charge shifts the optical transition from $C_0$ to $C_s$ by $\Delta_s(r)$ depending on its distance $r$. Additionally, an ensemble of remote fluctuating charges broadens the signal from $C_s$ to $C_{s,b}$, depending on the charge density $\rho_{trap}$. **c** Relative electric field sensitivity $|\Delta E|/E$ to electric field changes as a function of electric field $E_s$ and trap density $\rho_{trap}$. Left of the white dashed line, Stark-shifts are not large enough to be resolved by the Rayleigh criterion. Larger field strengths are correlated with larger inhomogeneous broadening. Eq. (1) assumes $\Delta\mu = 6.1 \times 10^{-4}$ GHz/(MV/m)$^2$, $\Delta\alpha = -5.1 \times 10^{-5}$ GHz/(MV/m)$^2$, $\Delta\beta = -5.5 \times 10^{-8}$ GHz/(MV/m)$^3$ and $\Delta\gamma = -2.2 \times 10^{-10}$ GHz/(MV/m)$^4$[35]. **d** The sensor's resolution in determining the distance of an elementary charge based on differentiating two distinct charge traps as a function of the charge trap density and distances. The resolution was determined for a trap with variable distance $r$ and a bias field equivalent to a trap distance of 0.8 nm. The dashed white lines indicate the inscribed resolution thresholds.

and therefore does not require magnetic resonance methods[5]. Measuring the energy shift reveals the magnitude of the electric field at the sensor probe $E_s$ via the DC-Stark-shift

$$\Delta_{Stark} = -\Delta\mu E_s - \frac{1}{2}\Delta\alpha E_s^2 - \frac{1}{3!}\Delta\beta E_s^3 - \frac{1}{4!}\Delta\gamma E_s^4, \quad (1)$$

with $\Delta\mu$ the change in the dipole moment and $\Delta\alpha$, $\Delta\beta$, and $\Delta\gamma$ the differences between the higher-order polarizabilities[35]. Importantly, in contrast to non-inversion symmetric configurations of color centers, such as the nitrogen-vacancy center in diamond[36] and the silicon-vacancy center in silicon carbide[37], the negligible linear- and strong non-linear response due to inversion symmetry makes the sensor applicable to typical semiconductor dopant and defect densities. If $\Delta\alpha$ dominates and the observed $\Delta_{Stark}$ arises from a localized elementary charge $e$ at a distance $r$ from the sensor, then

$$\Delta_{Stark}(r) \sim \Delta\alpha/r^4. \quad (2)$$

Decreasing distances of charges to the sensor causes increasingly higher spectral shifts. This characteristic makes sensors that have an inversion center remarkably sensitive to charges in close proximity and insensitive to electric field background noise.

The ~$10^{-7}$ relative electric field sensitivity (Fig. 1c) allows for a spectral sensor read-out that provides exceptionally high spatial resolution, reaching down to a few Ångström, even for charge densities up to 100 ppm (Fig. 1d).

In this work, the SnV local probe is stationarily located inside a bulk crystal housed in a cryostat at 4 K, however, it could also be integrated into the tip of a scanning probe microscope[38] for position dependent measurements well established in magnetometry[39,40] or into a nanodiamond for integration with other materials[41] or even biological samples[42]. Alternatively to the SnV, also other $D_{3d}$ symmetric defects, such as the silicon-[29] or germanium-vacancy[30,43] and other inversion symmetric defects in other materials, for example, in silicon[44], could be applied as a local probe. To demonstrate the non-linear sensor principle, we utilize a single SnV that was created upon ion implantation and annealing.

## Results

### Determining charge trap positions at the atomic lattice scale

The electrometer and its environment are depicted in Fig. 2a. To demonstrate its sensing capability, we analyze the time-varying quasi-static electric field that is caused by charging and neutralization of crystal defects in the surrounding lattice under laser irradiation[45,46]. Using the recorded field magnitude at different configurations of the charge distribution, we can resolve the surrounding crystal defects at the lattice scale.

If all traps are neutral, the total field at the position of the probe is zero, and the optical transition of the SnV is unperturbed. A charged trap will induce an electric field $\vec{E}_s$, which Stark-shifts the optical transition energy according to Eq. (1). If a single elementary charge is located in proximity to the probe, the C transition is shifted more than its own linewidth, producing a spectral jump. (Fig. 1b). The magnitude of the spectral shift can be determined by comparing with the unperturbed case; adding both resonances in one spectrum leads to a unique optical fingerprint with two peaks.

To collect the spectra, we measure the fluorescence from the C transition of the sensor S1 under photoluminescence excitation (PLE) with a narrowband laser (Fig. 2b). The experiment is conducted under 0.5 nW (a power much below the expected saturation power[47] for excluding power broadening effects) of 619 nm resonant light. Additionally, a 2 μW (CW power) 450 nm 4 ms pulse is applied between line scans for SnV charge state reinitialization. From the spectral shift, we extract the charge-induced electric field. Knowing the local field and using the polarizability enables us to estimate the trap-probe distance.

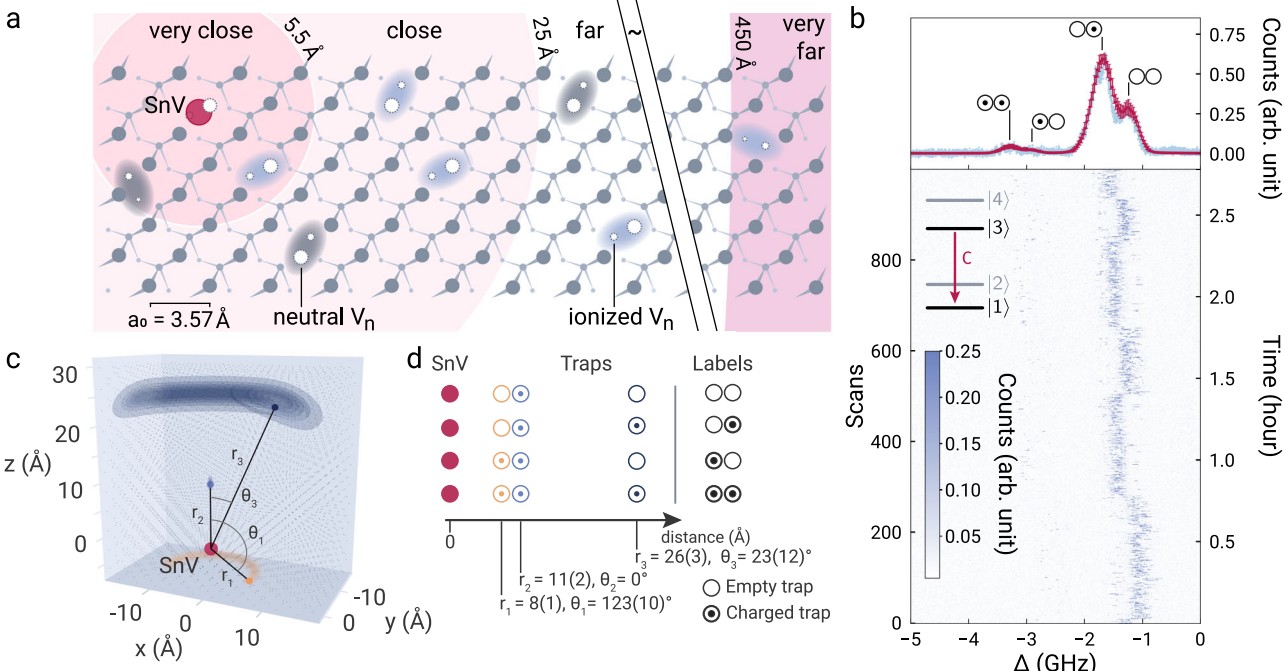

**Fig. 2 | Lattice scale localization of charge traps. a** Illustration of a diamond lattice including SnV and defects, identified as multi-vacancy complexes ($V_n$). Charges localized in these traps induce a Stark-shift of the energy levels of the atomic sensor probe. From very close to far, respectively, the spectral impact of an elementary charge can be categorized as follows: Spectral shifts larger than 30 GHz are detectable by photoluminescence spectroscopy, spectral shifts of -GHz are detectable by photoluminescence excitation spectroscopy (PLE), and inhomogeneous broadenings are detectable by PLE. Charges in the very far region have negligible effects. **b** Top: An integrated multimodal PLE spectrum recorded with

the SnV sensor S1 (blue) and modeled with Monte-Carlo simulations (red, error bars represent statistical standard deviation) to identify a proximity charge trap configuration (states labeled above the peaks, ⊙ and ○ represent ionized and neutral traps, respectively) and the surrounding charge density. Bottom: Time-resolved individual linescans. Inset: SnV level scheme and the probed transition. **c** Left: Identified charge trap configuration corresponding to (**d**), their relative positions and their probability distribution with respect to the SnV probe. Distributions resemble a donut shape due to the direction-independent calibration of the sensor. RIGHT: Table indicating the charge states and position of the identified traps.

For $N$ charged traps in the probe's vicinity, the electric fields add up to $\vec{E}'_s$ and the individual charges cannot directly be separated. To distinguish the $2^N$ charge states, the Stark-shifted PLE spectra are recorded repeatedly. Due to laser irradiation, the traps will be ionized and neutralized randomly. By sampling a large set of configurations complex trap distributions can be analyzed.

In addition to nearby charges that cause significant spectral line shifts, the numerous randomly distributed traps in the distant surroundings also contribute. These traps exhibit fluctuating charge states, resulting in a fluctuating electric field $\delta\vec{E}_s$ that causes inhomogeneous broadening. Consequently, the density of charge traps $\rho_{trap}$ within the lattice can be determined using linewidth measurements. We find that traps can be resolved with subnanometer resolution. For trap densities $\rho_{trap} \approx 0.3$ ppm, detection volumes of $150^3\,\text{Å}^3$ are feasible. Fluctuating charge traps at larger distances primarily contribute to inhomogeneous broadening.

Finally, to fully calibrate the electrometer, we consider its nonlinear response to external fields, causing an interdependence of the different external field components. For example, the effective Stark shift induced by two charges does not equal their sum. This phenomenon enables high resolution but makes the analysis of recorded fingerprints highly complex. We therefore build a theoretical database of simulated spectra for a large variety of discrete proximity charge positions and remote trap densities using Eq. (1).

We now analyze the complex experimental four-peak fingerprint from Fig. 2b quantitatively. We use experimentally determined polarizabilities[35]. By comparing experimental and simulated fingerprints, we find several possible trap configurations. From these

possible configurations, we identify the most plausible by specific physical considerations (Supplementary Fig. 3).

The most likely configuration of nearby traps consists of a permanent $\mathbf{E}_{bias}$, generated, for example, by a permanently ionized trap, and two additional traps inducing spectral jumps. We assign labels to the spectral peaks in Fig. 2b based on the charge state of the two additional proximity traps {○○, ○⊙, ⊙○, ⊙⊙}, where ○ signifies an uncharged trap, and ⊙ a charged trap. Subsequently, we determine the position of these charge traps up to an azimuthal angle using Monte-Carlo simulations. We extract the relative Stark-shifts corresponding to the proximity trap distances $r_1 = 8(1)$ Å, $r_2 = 11(2)$ Å, $r_3 = 26(3)$ Å (Fig. 2c, d) and a remote trap density of 74(22) ppm. Furthermore, we find for two charges, azimuthal angles with $\theta_1 = 123(10)°$ and $\theta_3 = 23(12)°$. The polar angles cannot be further confined because of the cylindrical symmetry of the problem setup.

## Charge dynamics
For identifying the positions of charge traps, we have used accumulated spectral fingerprints that reflect the integrated spectrum for the entire set of charge states. However, understanding time-resolved charge transfer dynamics requires comparing individual read-out events of our electrometer and, therefore, single PLE line-scans. We interpret the charge state changes with a simplified charge transfer picture (Fig. 3a): Charge traps, later identified as multi-vacancy complexes $V_n$, can be ionized under laser illumination through two distinct processes: negative charging, which occurs when the trap captures an electron promoted from the valence band leaving a positively-charged hole in the band; and positive charging

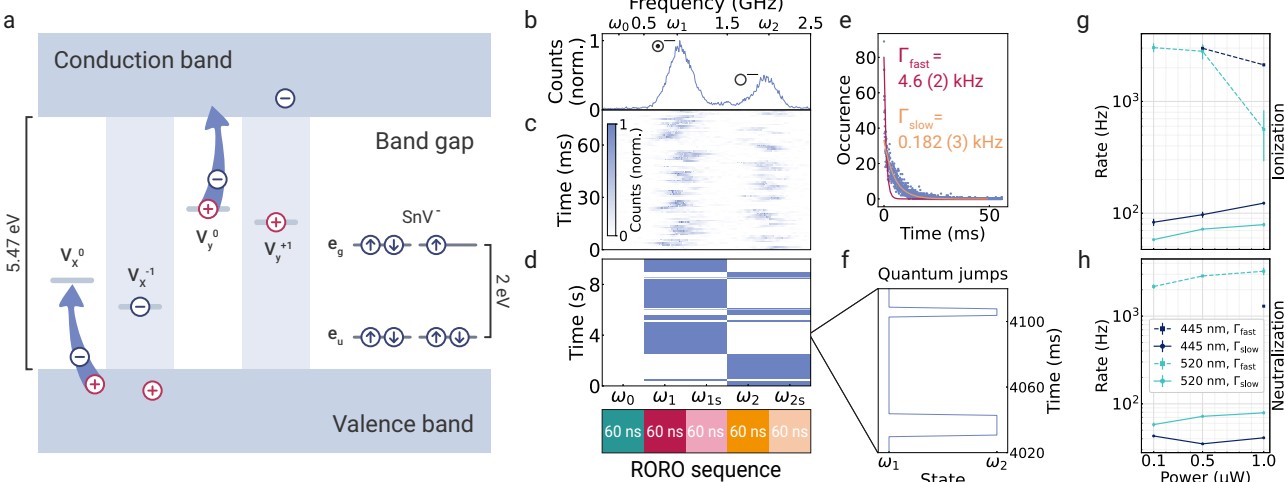

**Fig. 3 | Nanoscale charge dynamics in diamond. a** Illustration of an SnV, ionized ($V_i^{\pm 1}$) and neutral ($V_i^0$) defects, i.e., the charge traps, in the diamond bandgap. Ionization occurs when either an electron is promoted from the valence band to the charge trap or an electron is promoted from the trap to the conduction band by laser irradiation. Neutralization occurs when the trap either catches a hole from the valence band or an electron from the conduction band. **b** Integrated PLE spectrum and **c** time-resolved linescans acquired by the SnV sensor S2. **d** Using the specific resonance for each charge state obtained from the PLE spectrum, the laser frequency is periodically switched between the off-resonant $\omega_0$ and the resonances $\omega_1$, $\omega_2$. Each resonance is probed in two 60 ns steps with an additional frequency offset to compensate for spectral diffusion. The entire rapid optical readout (RORO) sequence takes 300 ns. Counts registered during each acquisition step are analyzed to determine how long a particular resonance remains bright. **e** An example histogram of the bright times of the resonance $\omega_1$ before a spectral jump to $\omega_2$

occurred. The data is fitted with a bi-exponential function that has two rates with constants $\Gamma_{\text{fast}}$ and $\Gamma_{\text{slow}}$. **f** A zoom-in of the spectral jumping plot from subfigure d reveals fast quantum jumps occurring on the order of ms. **g** and **h** To investigate the influence of photon irradiation energies on the recharging dynamics, spectral jumping rates are acquired under distinct laser wavelengths and powers. Rates are extracted separately for the investigated nearby trap's spectral jumping directions corresponding to ionization (redshift) and neutralization (blueshift) processes. Dark blue data points are acquired under 445 nm and green data points under 520 nm laser illumination. For each wavelength, the measurement is repeated at three different powers. Data points connected by solid lines correspond to the extracted $\Gamma_{\text{slow}}$ exponential rate constant. Fast rates $\Gamma_{\text{fast}}$ are connected by dashed lines and shown for histograms in which a fit to a bi-exponential distribution was possible. The error bars represent 95% confidence intervals extracted from the fits.

when an electron is promoted from the trap to the conduction band. Generated holes and promoted electrons then diffuse and recombine with other charge traps, leading to an overall charge-neutral environment. We denote the event $\bigcirc \rightarrow \odot$ as ionization and the inverse $\odot \rightarrow \bigcirc$ as neutralization event. The charge transfer picture[4,48] is consistent with the autocorrelation measurements we performed on the sensor emission (Supplementary Fig. 6).

To showcase the characterization of a local charge environment and dynamics of the charge transfer processes, we acquire a high timing-resolution set of data from a second sensor S2. The temporal analysis of the data acquired from S1 is provided in the supplementary materials. S2's charge dynamics are dominated by a single spectral jump, as shown in Fig. 3b. We label the two states as the neutral $\bigcirc$ and ionized $\odot$ states following our previous description.

The individual lines in Fig. 3c are acquired with a GHz/μs scanning speed by using a chirped frequency modulation of an EOM-generated sideband of a narrowband laser.

After identifying the resonance peaks in the integrated spectrum using the chirped frequency modulation, we perform a rapid optical read-out (RORO) of the charge trap (Fig. 3d). We generate a readout sequence by repeatedly modulating the sideband targeting five frequencies with 60 ns acquisition steps: off-resonant $\omega_0$, the peaks of two resonances $\omega_{1,2}$ and some pre-selected frequency offset $\omega_{1s,2s}$ to probe the resonances even if some spectral diffusion has taken place. The entire RORO sequence lasts a total of 300 ns. Using a single-shot read-out sequence, we extract a specific charge state of the sensor. We then visualize the distribution of duration the sensor stayed on one of the resonances in a histogram, as exemplified for the first resonance $\omega_1$ in Fig. 3e. The histograms are fitted with a bi-exponential distribution featuring two charge transfer rates $\Gamma_{\text{fast}}$ and $\Gamma_{\text{slow}}$, reflecting slow and fast charge dynamics as detailed below.

To analyze the charge dynamics on the shortest time scale available in the specific measurement configuration, we zoom into the data of Fig. 3d and look closely at the quantum jumps (as shown in Fig. 3f). We observe that although a time period can look bright on the scale of seconds in Fig. 3d, faster jumps occur on the millisecond scale. We further investigate the source of the bi-stability of the rates by selectively analyzing windows from an extended dataset acquired under the same conditions as in Fig. 3d. We find that if a time interval is predominantly bright for one resonance, the jumping rate from that state is closer to the extracted slow rate. For example, a predominantly bright time window of a total of ~31 s for $\omega_1$ yields a jumping rate $\Gamma_{\omega_1 \rightarrow \omega_2} = 26(5)$ Hz. On the other hand, when the time interval seems mainly dark, the jumping rate decays quicker, e.g., during a predominantly dark time window of ~13 s for $\omega_1$ yields $\Gamma_{\omega_1 \rightarrow \omega_2} = 1.91(13)$ kHz. While identifying the underlying physical process of the rate change requires further investigation, one possible explanation could be the change of a distant charge donor's trap state. The rate of the spectral jump may be modified by the availability and quantity of the required free charged particles.

To further experimentally investigate the origin of the charge dynamics, we repeat the measurement with a second laser applied during the off-resonant part of the readout sequence to characterize the charge dynamics induced by the properties of the illumination. Ultimately, such a measurement allows us to spectroscopically investigate the local charge environment and, more specifically, how its dynamics are influenced by external laser irradiation or other sources of noise in the nanoscopic local environment. We investigate six conditions: for both 445 and 520 nm wavelengths, three laser power levels are set. We plot the extracted rates for ionization and neutralization processes, respectively, in Fig. 3g and h.

An interesting and at first sight unexpected observation is made that the neutralization process is observed. Green illumination induces

faster jumps than the higher energy blue illumination. This may be caused by a two-photon process involving a defect level within the bandgap that is more likely to interact with the green laser than direct ionization from or to the valence and conduction bands. Acquiring data at additional power levels, illumination wavelengths and conducting measurements on different emitters with varying trap environments would enable extending this proof-of-principle experiment to a more detailed study.

Given the exceptional timing resolution of our RORO method, we can determine charge processes exceeding kHz rates, which are orders of magnitude faster than the typical ~1 Hz modulation rates of tunable lasers. Using conventional PLE by applying a voltage signal is not quick enough to observe the charge dynamics in the sample. Moreover, RORO could reveal charge processes with several MHz rates by utilizing the 60 ns minimum read-out steps. Reaching this limit requires estimated photon detection rates of ~17 MHz. By further reducing emitter lifetimes, for example through the Purcell-enhancement by cavity integration, the 60 ns read-out resolution could be reduced to several nanoseconds (see Supplementary Table I).

### Evaluating spectral diffusion

The charge transfer-induced spectral dynamics are highly detrimental for quantum technological applications. Spectral diffusion, a term indicating the probabilistic nature of the observed spectral dynamics, leads to optical decoherence[23–26], which results in reduced entanglement fidelity in quantum network nodes[27,49].

Knowing the non-linear receptiveness of our quantum electrometer to charge-noise, we now make predictions of how a specific charge distribution influences the spectral properties of a color center. Based on our model, we provide an overview of the inhomogeneous broadening caused by a certain charge trap density $\rho_{trap}$. The details of the calculation are provided in supplementary information.

We first focus on the bulk case (Fig. 4a) and then analyze surface charge traps ($\rho_{trap}^s$) for two different surface geometries, planar (Fig. 4b) and cylindrical (Fig. 4c). We find that an implantation depth of $d > 21$ nm and a cylinder with a radius of $r > 45$ nm will warrant that surface charges do not deteriorate the spectral properties of an SnV color center with linewidth broadening of <1%. Such broadening leads to 90% interference visibility and more than 87% entanglement fidelity[49]. Similar estimations can be performed for any defect given a known polarizability. Control measurements concerning spectral diffusion with respect to the illumination field are provided in the supplementary information.

The estimated minimum detrimental distances make SnVs and similar color centers well suited for the integration into nanostructures that enhance photon collection efficiencies[50] and provide tailored emission properties via the Purcell-effect[51,52] for quantum information applications.

It must be noted that an increase in the absolute lifetime limited linewidth due to the Purcell effect would decrease the depth and radius criteria. Furthermore, these analyses do not take band-bending and Fermi level modification effects into account, which could have additional implications on the charge state stability and brightness of the color centers.

### Identifying material properties: divacancy formation

Based on the electrometers ability to quantify charge trap densities, we extend our investigation of material properties and combine our sensor data with additional simulations to answer further open research questions. Specifically, we determine the physical origin of charge traps in implanted diamond. Considering a sample with <1 ppb of nitrogen and boron, and even lower lattice defect concentrations[53], the estimated charge trap density of 74(22) ppm must originate from the Sn-ion implantation damage and the

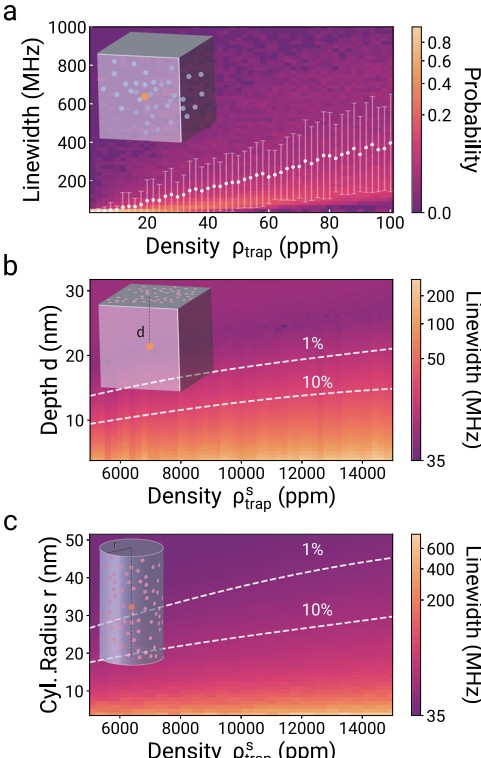

**Fig. 4 | Simulations of the SnV zero phonon line inhomogeneous broadening due to bulk and surface charges. a** Inhomogeneous broadening of an SnV with a lifetime-limited linewidth of 35 MHz as a function of $\rho_{trap}$ in units of ppm in bulk diamond. The colors indicate the linewidth distribution. The mean and variance of the distribution are shown by the white dots and error bars. **b** Inhomogeneous broadening as a function of the distance of an SnV to a planar surface and the surface trap density $\rho_{trap}^s$ defined as a fraction of surface lattice sites. The dashed lines show the threshold of 1% and 10% of broadening compared to the lifetime-limited linewidth of 35 MHz. **c** Inhomogeneous broadening of an SnV centrally located in a cylinder with radius $r$ as a function of $\rho_{trap}^s$. The dashed lines signify 1% and 10% broadening.

subsequent annealing process. Implantation of ions produces Frenkel pairs: a pair of one monovacancy $V_1$ and one dislocated interstitial carbon atom. During annealing $V_1$ becomes mobile and can form vacancy complexes, a process not well understood and an active area of research[54–56] (Fig. 5a).

Here, we estimate the $V_1$ to divacancy $V_2$ conversion yield using a kinetic Monte-Carlo simulation[57] in combination with a simple stochastic diffusion model described in the supplementary information. We consider the $V_2$ density as a proxy for higher-order vacancy complexes $V_n$. Annealing up to 1100 °C primarily converts $V_2$ into $V_3$ and $V_4$[54,58]. We indeed observe wavelength-dependent spectral diffusion and jumps (Supplementary Fig. 10), indicating different ionization energies of the multiple trap species. We interpret the estimated density of $V_2$ as both an order-of-magnitude approximation and an upper limit for the overall charge trap density. We estimate and compare $\rho_{V_2} = 40.0(2.1)$ ppm to the experimentally estimated trap density of $\rho_{Exp} = 74.1(22.5)$ ppm. Due to charge neutrality, the overall charge density $\rho_{Sim}$ would correspond to twice the density of $V_2$ with $\rho_{Sim} = \rho_{V_2} \times 2 = 80.0(4.2)$ ppm. We attribute the small mismatch to a reduction in the density of the $V_n$ compared to the $V_2$ estimation.

Understanding the origin of the charge traps also provides a clear path on how to create optically noise free group-4 vacancy defects in diamond. Single-peak fingerprints, indicating a low $V_n$ density, are more frequently observed in high-pressure high-temperature (HPHT)

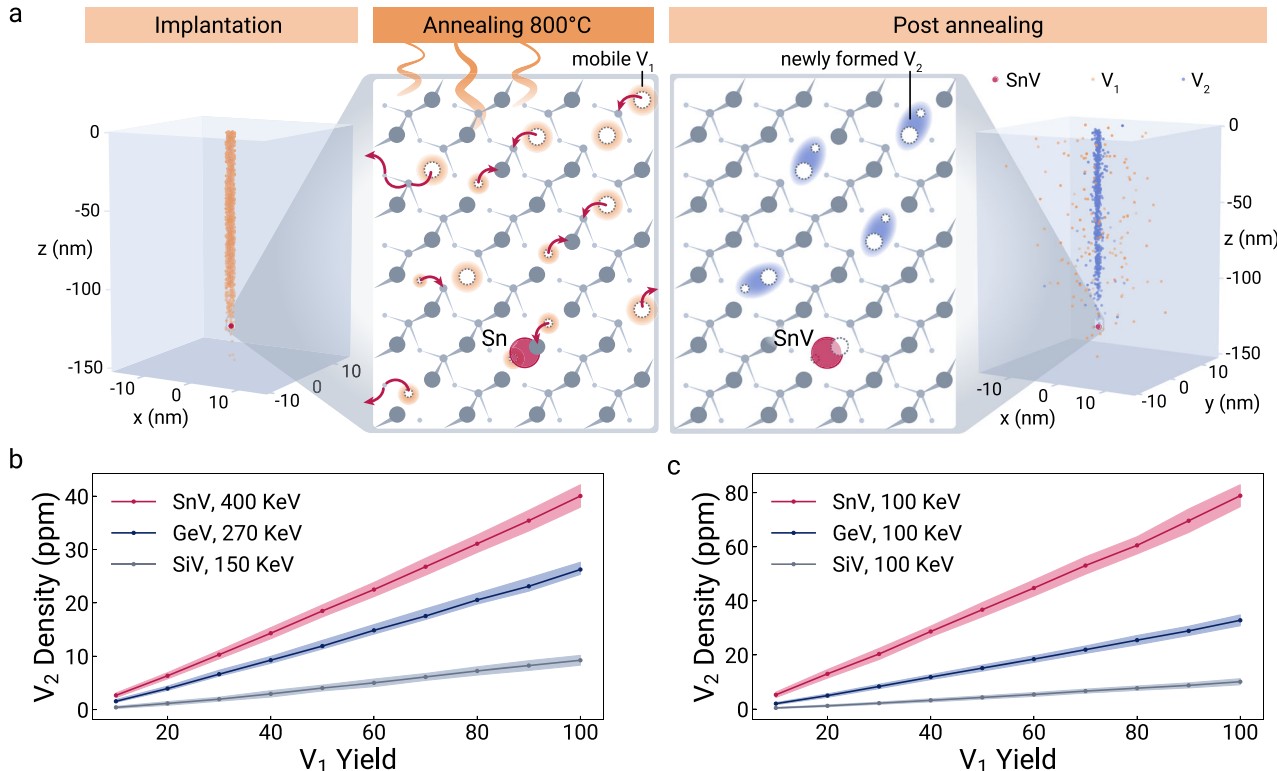

**Fig. 5 | Simulations of divacancy ($V_2$) creation during annealing. a** Implantation panel: Spatial distribution of created monovacancies ($V_1$) by 400 keV Sn implantation predicted by SRIM[67] simulations. Annealing panel: $V_2$ formation during annealing at 800 °C. At elevated temperatures, $V_1$ start diffusing. Then, $V_1$ immobilize either by moving to the boundaries, recombining with interstitial carbons, or forming $V_2$. Post annealing panel: Distribution of $V_1$ and $V_2$, which are distributed in the vicinity of the damage channel caused by the Sn implantation. Zoom in: $V_2$ in the vicinity of the SnV. **b** Simulated densities of $V_2$ as a function of the $V_1$ implantation yield (% of participating $V_1$ estimated by SRIM simulations) for the three different species Sn, Ge, and Si. A reduced $V_1$ yield is attributed to recombination with interstitial carbons. The implantation energies are selected so that an average implantation depth of 100 nm is reached. **c** Densities of $V_2$ for 100 keV implantation energy. The error bands represent the statistical standard deviation.

annealed samples at 2000 °C[48], which is in agreement with electron spin resonance measurements[58].

Spectral jumps have been reported before for group-4 vacancy defects[47,59,60]. Comparing the $V_n$ density for the atomic species Si, Ge, and Sn and varying implantation energies (Fig. 5b, c) indicates that Si implantation leads to the lowest $V_2$ density. This observation is in accordance with the more frequent reports of spectrally stable SiV[61] compared to SnV, now explained with our analysis that heavier ions cause increasing $V_n$ densities.

## Discussion

In conclusion, we use the SnV in diamond, a representative of an inversion symmetric point defect in wide-bandgap materials, as a local probe to detect localized charges with Angstrom resolution at the lattice scale. For this work, we constrained some degrees of freedom in the positions of the traps for the Monte Carlo simulation to make it more time-efficient. The main constraints were the confinement of the remote charges to a conical volume $z > z_0$, where $z_0$ is the assumed implantation depth of the SnV. The conical volume modeled the lattice damage remaining after annealing. Furthermore, we confined the three charge traps responsible for the multimodal spectrum to a plane. These limitations are not fundamental to this method and can be relaxed. In principle, a more extensive parameter space can be explored, but this would require significantly longer simulation times and increased memory demands. The addition of a more in-depth calibration of the probe would ameliorate this shortcoming. Particularly, implementing a probe-specific multi-directional calibration process, with the inclusion of extracting direction-

dependent polarizability coefficients, would allow us to construct suitable polarizability tensors. Using such a calibrated probe in combination with the Monte Carlo simulations would help enable azimuthal localization for full positioning. Alternatively, correlated sensing[15,16] can be used to reduce the uncertainty of the charge traps' absolute position in our work.

We utilize a rapid spectroscopy technique based on frequency modulation with electro-optical modulators, to enable time-resolved access for recording and measuring charge dynamics of single non-fluorescent defects under laser irradiation with MHz readout rates, demonstrating a time-resolved quantum electrometer working at the atomic scale (see Table 1). Such observations on the single-charge scale open the possibility to further understand the origin and type of charge defects and charge transport phenomena. Intriguingly, a sensor such as the one proposed in this study can be used to study topological quantum phenomena of ferroelectrics, for example, the detection of ferroelectric vortices or polar skyrmions[62].

An exciting secondary application is to use our sensor for sub-diffraction resolution position estimation. The electrometer's sensitivity to background charge-noise can be used to sense the position of an illumination laser with sub-diffraction precision. We estimate that a resolution below 1 nm can be achieved (Supplementary Fig. 8).

From the analysis of the local charge environment, we are able to understand the nanoscopic origin of spectral diffusion of SnVs and formulate mitigation strategies. We identify that the local defect density of $V_n$ should be reduced, and quantify precisely the maximally allowed charge trap density for reaching optical coherence.

**Table 1 | Summary of previously developed electrometers or related works**

| Reference | Platform | Single charge localization | Precision | Sensitivity in V/ m/√Hz | Temporal analysis | Based on | Operating temperature |
|---|---|---|---|---|---|---|---|
| 5 | NV | ✓ | ~nm | × | × | MW polarization-dependent ODMR | RT |
| 6 | QD | ✓ | × | 5 (DC), 140 (AC) | × | Modulation spectroscopy | 4.2 K |
| 7 | NV | × | × | 891 (DC), 202 (AC) | × | FID (DC), Hahn-Echo (AC) | RT |
| 8 | NV | ✓ | 2 nm | × | × | Ramsey | RT |
| 9 | QD | ✓ | 70 nm volume | × | × | PLE, photon statistics | 5 K |
| 10 | NV | × | × | × | × | ODMR | RT |
| 11 | NV | × | × | × | × | ODMR | RT |
| 12 | NV | × | × | × | × | CDD | RT |
| 13 | NV | × | × | × | × | Qdyne | RT |
| 14 | NV | × | × | $26 \times 10^3$ (AC) | × | Ramsey lock-in (DC) XY4 lock-in (AC) | RT |
| 15 | NV | ✓ | 1.6 nm | × | ✓, ~1 Hz | PLE, correlated analysis | 9 K |
| 16 | NV | ✓ | 1.7 nm | × | ✓, ~1 Hz | PLE, correlated analysis | 11 K |
| 43 | GeV | ✓ | × | × | ✓, ~1 Hz | PLE | 4 K |
| 70 | NV | × | × | × | ✓, 1 kHz | PL | RT |
| Our work | SnV | ✓ | 1;Å | × | ✓, ~MHz | PLE, MCS, RORO | 4 K |

'Ramsey, FID, CDD' and 'XY4, Hahn-Echo, Qdyne' are MW spin control-based phase-sensitive methods suitable for DC and AC measurements, respectively.
*NV* nitrogen-vacancy center, *RT* room temperature, *MW* microwave, *QD* quantum dot, *DC* direct current, *AC* alternating current, *FID* free induction decay, *PLE* photoluminescence excitation, *CDD* continuous dynamical decoupling, *ODMR* optically detected magnetic resonance, *GeV* germanium-vacancy center, *PL* photoluminescence, *SnV* tin-vacancy center, *MCS* Monte-Carlo simulations, *RORO* rapid optical readout.

Building on the insights of our work, we believe that our electrometer opens up an exciting direction in material science, enabling the time-resolved study of single elementary charges with Ångström spatial and 100 ns temporal resolution.

The integration of the sensor into a scanning-probe tip or a nanodiamond will allow for the study of single and multiple lattice defects in silicon transistors, optically active quantum memories, and defect-induced charge-noise in on-chip ion and superconducting computers, potentially mitigating these detrimental effects and contributing to the further optimization of materials for the application in quantum technology.

## Methods
### Monte Carlo simulations
Here, we provide an overview of the general methodology of simulating single and multimodal spectra (Fig. 2b). The simulations begin with distributing charge traps uniformly within a specified volume or surface. For multimodal spectra, such as the one depicted in Fig. 2b, the distribution of charge traps is divided into two categories: proximity traps and remote traps. Proximity traps are positioned at fixed locations, while remote traps are distributed with a fixed density within a prescribed volume. The $SnV^{-1}$ is always located at the origin of the coordinate space.

Once a spatial trap configuration is created, a single iteration of the Monte Carlo simulation can be performed. It consists of assigning charges to the trap locations (charging of traps). A fixed number of charges are distributed assuming charge neutrality $-e + e\sum_i q_i = 0$, where $e$ is the elementary charge and $q_i$ a charge state $q_i \in \{-1, 0, +1\}$. The field strength at the location of the $SnV^{-1}$ then becomes:

$$\mathbf{E} = \sum_i \mathbf{E}(q_i, \mathbf{r}_i), \tag{3}$$

where $\mathbf{r}_i$ is the position of a trap and $\mathbf{E}(q_i, \mathbf{r}_i)$ is the electric field of a point charge in the medium, chosen such that it adequately reflects boundary conditions for the solution of Maxwell's equations. The non-linear Stark shift $\Delta_{\text{Stark}}$ corresponding to the magnitude of the field $\mathbf{E}(q, \mathbf{r}_i)$ is calculated using Eq. (1), where $E_s = |\mathbf{E}_s|$ and the parameters $\Delta\mu = 6.1 \times 10^{-4}$ GHz/(MV/m)$^2$, $\Delta\alpha = -5.1 \times 10^{-5}$ GHz/(MV/m)$^2$, $\Delta\beta = -5.5 \times 10^{-8}$ GHz/(MV/m)$^3$ and $\Delta\gamma = -2.2 \times 10^{-10}$ GHz/(MV/m)$^4$ [35]. Unless explicitly stated otherwise, the procedure is repeated 1000 times and the spectrum corresponding to the distribution of Stark shifts is generated using

$$S(\omega) = \frac{1}{N} \sum_n L_\gamma(\omega - \Delta_{\text{Stark}, n}), \tag{4}$$

where $N$ is a normalization constant (max $S(\omega) = 1$), $n$ is the simulation step index and $L_\gamma(\omega)$ is a Lorentzian line profile with full width half maximum $\gamma = 35$ MHz corresponding to the lifetime-limited linewidth of the $SnV^{-1}$ [47]. We assume that there is no additional power-broadening, nor a lifetime reduction due to Purcell enhancement.

The process of determining the most probable trap configuration is divided into three main steps. First, we utilize the four peaks observed in the measured multimodal spectrum to identify the positions of proximity traps that generate Stark shifts consistent with the experimental observations. This initial step provides a rough estimation of the proximity trap positions. Next, we employ an optimization procedure to fine-tune the predetermined positions of the proximity traps. By optimizing the relevant parameters, we generate a comprehensive database of simulated spectra. Finally, utilizing the objective function ($\chi^2$ test) employed during the optimization procedure, we analyze the large dataset of simulated spectra to identify the most likely proximity trap configuration. This objective function serves as a measure of the agreement between the simulated spectra and the experimental observations. By comparing the calculated spectra with the measured data, we can identify the configuration that best matches the experimental results. In the following, we provide a detailed description of each individual step.

The four peaks in the measured spectrum shown in Fig. 2b are used as a reference to estimate the location of a charged proximity trap relative to the SnV using Eq. (1).

We deem the scenario of three traps contributing to the multimodal spectrum the most likely due to the temporal analysis of the PLE data (see Supplementary Information). The trap situated at

$\mathbf{r}_2 = (0, 0, r_2)$ is assumed to be permanently charged. The charge state of the other two traps is then given by $\{\bigcirc\bigcirc, \bigcirc\odot, \odot\bigcirc, \odot\odot\}$, where the left circle represents a trap at position $\mathbf{r}_1$ and the right circle a trap at position $\mathbf{r}_3$. An empty circle represents a neutral trap, whereas a filled circle marks a trap with a charge. The peaks corresponding to each charge state are shown in Fig. 2b. The negative charge located at a distance $\mathbf{r}_2$ magnifies the response of the SnV to remote charges and produces the observed inhomogeneous broadening. The choice $\mathbf{r}_2 = (0, 0, r_2)$ is, of course, not the most general, but we select it due to the anisotropy expected from the implantation procedure. Furthermore, including the position $\mathbf{r}_2$ with two more degrees of freedom would have made the free parameter space too large and increased the computational time.

We limit the placement of the three proximity traps to a plane, therefore further reducing the complexity of the problem. We approximate the initial positions $\mathbf{r}_1$, $\mathbf{r}_2$, $\mathbf{r}_3$ by solving the simultaneous set of equations:

$$\Delta_{\bigcirc\bigcirc} = -\frac{\alpha^2}{2} E(-1, \mathbf{r}_2)^2 \tag{5}$$

$$\Delta_{\bigcirc\odot} = -\frac{\alpha^2}{2} [E(-1, \mathbf{r}_2) + E(-1, \mathbf{r}_1)]^2 \tag{6}$$

$$\Delta_{\odot\bigcirc} = -\frac{\alpha^2}{2} [E(-1, \mathbf{r}_2) + E(-1, \mathbf{r}_3)]^2 \tag{7}$$

$$\Delta_{\odot\odot} = -\frac{\alpha^2}{2} E(-1, \mathbf{r}_2) + E(-1, \mathbf{r}_1) + E(-1, \mathbf{r}_3)]^2 \tag{8}$$

The positions are parameterized according to

$$\mathbf{r}_1 = r_1[\cos(\theta_1), 0, \sin(\theta_1)] \tag{9}$$

$$\mathbf{r}_3 = r_3[\cos(\theta_3), 0, \sin(\theta_3)] \tag{10}$$

The above equations can be solved such that $r_1(\theta_1)$, $r_3(\theta_1)$ and $\theta_3(\theta_1)$. The relative shifts $\Delta_{\bigcirc\odot}$, $\Delta_{\odot\bigcirc}$, $\Delta_{\odot\odot}$ are estimated from the central peak positions using a fitting procedure, where the integrated spectrum in Fig. 2b is fitted simultaneously with four Voigt profiles. The choice of $\mathbf{r}_2 = (r_2 \sin(\theta_2), 0, r_2 \cos(\theta_2) r_2)$ was based on a coarse scan of the Monte Carlo procedure by varying the angle $\theta_2$. We found that in conjunction with the remote traps, $\theta_2 = 0$ produced the most consistent results.

The fine-tuning of the proximity trap positions in the second step is again done using a Monte Carlo simulation in combination with an optimization procedure. For the optimization procedure, remote traps are distributed randomly in a conical volume $z > 0$ nm with an opening angle of 45° at a fixed density $\rho_{\text{trap}}$, to mimic the non-isotropic distribution of traps, which is expected to result from implantation damage. The volume is capped at 30 nm. We exclude a spherical volume with a radius of $r_q < 2.5$ nm, for placing the proximity charges. A charged trap is assumed to contribute to the total field

$$\mathbf{E} = \sum_i \mathbf{E}(q_i, \mathbf{r}_i), \tag{11}$$

with

$$\mathbf{E}(q_i, \mathbf{r}) = \frac{q_i}{4\pi\epsilon_0\epsilon_r} \frac{\mathbf{r}}{r^3}, \tag{12}$$

where $e$ is the elementary charge and $q_i$ a charge state $q_i \in \{-1, 0, +1\}$, $\epsilon_0$ is the vacuum permittivity and $\epsilon_r = 5.5$ is the relative permittivity of diamond.

For each choice of $\rho_{\text{trap}}$, $r_2$ and $\theta_1$, we perform a Monte Carlo simulation of the spectral fingerprint as described in the overview of the methods section of the main text.

To adequately account for the charge state of the proximity traps, they are charged with a probability $p_i$ according to the relative peak heights in each individual step of the simulation. We use $p_{\bigcirc\bigcirc} = 0.31$, $p_{\bigcirc\odot} = 0.63$, $p_{\odot\bigcirc} = 0.017$ and $p_{\odot\odot} = 0.041$.

The optimization of the trap positions for a given $\rho_{\text{trap}}$, $r_2$ and $\theta_1$ is done by minimizing $\chi^2$

$$\sum_i \chi(\boldsymbol{\theta}, i) = \sum_{n=0, i}^N \frac{[O_n(\boldsymbol{\theta}, i) - E_{n,i}]^2}{E_n}, \tag{13}$$

with the simplicial homology global optimization (shgo) algorithm. We use an implementation of the shgo algorithm provided by the Python library *SciPy*[63]. In Eq. (13), $\boldsymbol{\theta} = [a, b]$, where $a$, $b$ fine tune $r_2' = ar_2$ and $r_3' = br_3$.

We split the spectrum into three parts associated to $i \in \{\bigcirc\bigcirc, \bigcirc\odot, \odot\bigcirc + \odot\odot\}$. For each part, we use the respective single and double Voigt profile fits for comparison with the simulated spectra by using Eq. (13). In Eq. (13) $E_{n,i}$ are the expected counts in the $n$th bin, which is found by binning the (normalized) single and double Voigt profiles fitted to the measured spectrum into 170 equally sized bins over an interval containing the profile with a width of 4 GHz. $O_n(\boldsymbol{\theta}, i)$ is number of expected counts of the respective $i$ for the simulated spectrum in the $n$th bin.

Finally, the values $\chi^2$, $r_1'$, $r_3'$ are then tabulated for $\rho_{\text{trap}} \in [35, 100]$ ppm, $\Delta_{\odot\odot} \in [0.5, 1.7]$ GHz and $\theta_1 \in [0, 0.6]$ rad. We perform 500 iterations of the optimization over distinct spatial configurations of the remote traps for each value of $\rho_{\text{trap}}$, $r_2$ and $\theta_1$. We only use the 50 lowest values of $\chi^2$ (the others are considered outliers), and perform a weighted average for determining $\langle\chi^2\rangle$.

We find the 68% confidence intervals for $\rho_{\text{trap}}$, $\Delta_{\odot\odot}$ and $\theta_1$ by $\min\{\langle\chi^2\rangle\} + 3.5$[64]. The results are shown in Fig. 2c, d. The statistical error shown in Fig. 2 is produced by all the simulated spectra within the 68% confidence interval.

## Samples

The samples used in this work (E001 for S1 and E013 for S2) are single-crystal electronic grade diamonds grown by chemical vapor deposition (supplied by Element Six Technologies Ltd. (UK) and with 100 faces). Both substrates are initially cleaned for about one hour in a boiling triacid solution ($H_2SO_4$:$HNO_3$:$HClO_4$, 1:1:1)[65] and then etched in $Cl_2$/He and $O_2$/$CF_4$ plasmas, in order to remove any organic contaminants and structural defects from the surface[66]. The diamond hosting S1 (E001) is then implanted with $^{120}$Sn (spin-0) ions, using a fluence of $5 \times 10^{10}$ atoms cm$^{-2}$ and an implantation energy of 400 keV, which corresponds to a penetration depth of 100 nm, as estimated by SRIM simulations[67]. The formation of the SnV color centers is finally achieved by low-pressure–high temperature (LPHT) annealing step at a temperature of 1050 °C for about 12 h in vacuum (pressure < $7.5 \times 10^{-8}$ mbar). The sample hosting S2 (E013) is instead initially implanted with sulfur ions, with a nominal fluence and implantation energy of $5 \times 10^{12}$ atoms cm$^{-2}$ and 160 keV, respectively. A LPHT annealing step ($T = 1050$ °C and $P \approx 1 \times 10^{-7}$ mbar), for about 12 h, is then used to heal the lattice damage caused by the implantation process. After a second triacid cleaning step, Sn ions are implanted in the same substrate, with nominal values $5 \times 10^{10}$ atoms cm$^{-2}$ and 400 keV for fluence and energy, respectively. According to SRIM simulations, the expected implantation depth is

about 100 nm for both S and Sn. After a second LPHT annealing step, the substrate is annealed at 2100 °C at $P \approx 6$–8 GPa for a total of 2 h and finally cleaned in a boiling triacid solution.

Nanopillars are fabricated on both samples by e-beam lithography and plasma etching. In the case of S1 (E001), 200 nm of $SiN_x$ were deposited on the surface of the diamond in an inductively coupled plasma (ICP) enhanced chemical vapor deposition system. After coating the sample with 300 nm of electro-sensitive resist (ZEP520A) and a few nm of a charge-dissipating layer (ESpacerTM), pillars with nominal diameters ranging from 180 to 340 nm (in steps of 40 nm) are exposed by means of electron-beam lithography. After development, the pattern was transferred into the $SiN_x$ layer by a reactive ion etching (RIE) plasma (10 sccm $CF_4$, RF power = 100 W, $P$ = 1 Pa) and then etched into the diamond during an ICP process in $O_2$ plasma (80 sccm, ICP power = 750 W, RF power = 200 W, $P$ = 0.3 Pa). The sample hosting S2 (E013) is coated with a similar stack of layers, but with the introduction of 10 nm of Ti between $SiN_x$ and resist to replace the ESpacerTM layer. Moreover, a different range of nominal diameters (from 140 to 260 nm, in steps of 20 nm) is used in this case for the patterned pillars. After development, the hard mask layer was etched by an ICP-RIE process in an F-based plasma ($SF_6$:Ar, 30:15 sccm, ICP power = 500 W, RF power = 35 W, $P$ = 1 Pa), followed by an ICP process in $O_2$ plasma (99 sccm, ICP power = 1000 W, RF power = 200 W, $P$ = 1 Pa) for the complete transfer of the pattern into the diamond substrate. The final height of the nanopillars is ~500 nm for both processed samples. The remaining nitride layer is finally removed in a solution of buffered HF, and the diamond surface is exposed.

### Measurement setup

The sample is cooled to 4 K in a closed-cycle helium cryostat (Montana s50). A home-built confocal scanning microscope is utilized to locate and optically address nanopillars with SnVs. PL spectra are measured via a spectrometer (Princeton Instruments HR500) with a CCD camera (Princeton Instruments Excelon ProEM:400BX3). Photons collected from the cryogenic setup are coupled into a fiber, converted into digital signals via avalanche photo diodes (APD, Excelitas SPCM-AQ4C or SPCM-AQRH) and counted via a data acquisition device (NIDAQ USB 6363).

Non-resonant confocal microscopy measurements are done with a green diode laser at 520 nm (DLnsec). The SnV's charge state is initialized by a blue diode laser (450 nm, Thorlabs LP450-SF15 or 445 nm Hübner Cobolt 06-MLD). Both green and blue lasers are employed for inducing charge dynamics in the investigated traps. For PLE scans and readouts, two 619 nm orange lasers, a tunable dye laser nm (Sirah Matisse, DCM in EPL/EG solution) and an SHG laser source (TOPTICA SHG DLC PRO) are employed. Orange lasers are used in the experiment by scanning the frequency of the resonant excitation laser across the C transition of an SnV center and collecting the phonon sideband of the fluorescence. The orange lasers' frequency is measured and stabilized using a wavemeter (High Finesse W7).

For the rapid PLE scans and optical readout sequences (RORO), a system based on sideband generation via an electro-optical modulator (EOM, Jenoptik AM635b) is utilized. When driven by a sinusoidal signal $\Omega_i$, the EOM creates two sidebands with frequency offsets $\omega_L \pm \Omega_i$, where $\omega_L$ is the laser's frequency. By chirping the modulation, it is possible to modulate the frequency of the sidebands for scanning or probing resonances. We generate the voltage signal using an arbitrary waveform generator (AWG, Keysight M8195A). We employ the blue-shifted sideband to excite the SnV.

We connect a pick-off path from the output of the EOM to the photodiode embedded into a modulation bias controller (MBC, OZ Optics Mini-MBC-1B0). MBC applies a bias voltage to the EOM and stabilizes it to its interferometric minimum. The MBC stabilization diminishes the central frequency component of the laser and any higher-order even sidebands. An optimized choice for the driving amplitude maximizes the first-order sideband.

The frequency-modulated sidebands are directed to the sample through the confocal microscope. Emitted photons from the SnVs are turned into digital signals via the APDs and timestamped (arrival time of a photon) using a timetagger (quTools qutag). A pulse streamer (Swabian Instruments 8/2) is used to trigger the AWG to start the sequence. It also generates a periodic calibration signal sent to one of the channels of the timetagger. A signal generator (Rigol DG952) streams a signal throughout the experiment to synchronize the clocks of the pulse streamer, timetagger, and AWG. Experiments are controlled with the software suite Qudi[68].

## Data availability

The data and code that support the findings of this study have been deposited in the Zenodo repository with https://doi.org/10.5281/zenodo.15704695[69].

## Code availability

The data and code that support the findings of this study have been deposited in the Zenodo repository with https://doi.org/10.5281/zenodo.15704695[69].

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

## Acknowledgements

The authors would like to thank Viviana Villafane for sample annealing, Alex Kühlberger for his support with implantation, Lilian Hughes for discussions regarding the formation of vacancy complexes in diamond during annealing, Özgün Ozan Nacitarhan and Kilian Unterguggenburger for their support on experimental control software, David Hunger, Boris Naydenov, Matthew Trusheim, Laura Orphal-Kobin and Julian Bopp for their feedback on the manuscript. Funding for this project was provided by the European Research Council (ERC, Starting Grant project QUREP, No. 851810, granted to T.S.), the German Federal Ministry of Education and Research (BMBF, project DiNOQuant, No. 13N14921, granted to T.S.; project QPIS, No. 16KISQ032K, granted to G.P. and T.S.; project QPIC-1, No. 13N15858, granted to T.S.), the Einstein Foundation Berlin (Einstein Research Unit on Quantum Devices, granted to T.S.) and the Alexander von Humboldt Foundation (granted to J.H.D.M.).

## Author contributions

G.P. was responsible for the modeling and executed a range of Monte-Carlo and stochastic simulations. C.G.T. constructed the confocal microscope under the guidance of J.H.D.M. and designed the overall experimental infrastructure and protocols. C.G. developed the EOM-based rapid optical readout setup. C.G.T. acquired the data from S1 and conducted the spectral diffusion and autocorrelation measurements. C.G. and C.G.T. collected the experimental data for S2, and C.G. analyzed the temporal data. F.M.H. performed the Rabi oscillation control measurements with a home-built SHG laser system, which she constructed. A.T. implanted the Sn atoms into the sample. T.P. led the sample preparation process and fabricated the nanopillars. T.S. developed the idea and supervised the project. G.P., C.G.T. and T.S. wrote the original manuscript with the input of all authors.

## Funding

## Competing interests

The authors declare the following competing interests: Patent applications with numbers DE 10 2024 003 454.4 and US 19/036,421 have been submitted by Humboldt-Universität zu Berlin, listing C.G.T., G.P. and T.S. as inventors. The patented aspects include the temporal analysis of the PLE data for identifying the charge trap configuration and the application of Monte Carlo simulations for localizing individual proximal charge traps and determining the density of remote traps. The applications are currently pending approval. The other authors declare that they have no competing interests.
