## [Transparent Peer Review file · Nature Communications]

Quantum Electrometer for Time-Resolved Material Science at the Atomic Lattice Scale

Corresponding Author: Professor Tim Schröder

Version 0:

Reviewer comments:

Reviewer #2

(Remarks to the Author)

It is good to see the authors refer to the recommended papers. However, spatially and temporally correlated sensing of local electric environment has been reported in many papers, besides Nat. Photon. 18, 230–235 (2024), see also Tom Delord, Richard Monge, and Carlos A. Meriles, Nano Letters 2024 24 (22), 6474-6479, which presents exact “simultaneous correlations in space and time” with NV system. There are neither relevant comments nor citations in the manuscript. Given the importance of the research area, there could be more pioneer works. I strongly suggest that the authors make systematic explorations in literature and comment on proper articles in the maintext. Again, the manuscript is not as claimed as “first time-resolved quantum electrometer working at the atomic scale”, neither the arXiv submission 2401.14290v1.

Besides the insufficient novelty, the main issue lies in the non-solid conclusion and unconvincing analysis in the simulation built on unreasonable assumptions, for example, limiting the placement of the three proximity traps to a plane. Why do the traps/charges have to be in a plane? Is it a specially patterned implanting? What is the precision in terms of scattering of the implantation in the x-y plane, is it better than the claimed location resolution? Same to the non-general choice of $r_2 = (0, 0, r_2)$. The assumptions are intensively made to simply the parameter space, but lead to unrealistic results, being lack of universality. Without the assumptions, there would be multiple solutions in the configuration of traps. Furthermore, how is the sign of r_2 determined?

As also pointed out, the authors should address in detail how come the full location of a charge in 3-dimension space with a single-point detector. It is trivial, a full location needs a radial distance, the polar angle, and the azimuthal angle, or three coordinates. In the section of “Multimodal spectra” in Supplementary Material, however, only a single parameter (θ_i , $i=1,3$, claimed as azimuthal angle) representing angle is found. How is the other determined? By the way, it is suggested that the authors mark out the angle in figures, e.g., fig. 2c, to avoid confusion.

The authors claimed that “no other effects contribute to the broadening” but failed to provide relevant experimental data. First of all, optical excitation power is a common factor inducing spectral broadening. It is necessary to provide the experimental test and make sure that there is no such broadening when determining the density of surrounding charges. In terms of the strain dynamics, the author referred to the work Ji et al, Nat. Photon. 18, 230–235 (2024). However, this is in general related to the quality of crystal lattice and can be hugely different from sample to sample, especially between the ones dealt with laser-writing (used in the reference) and those of implantation (this work).

In summary, I cannot recommend the publication before the above questions are well addressed. Without determinative solutions, it is unpersuasive to make statements of “full positioning” and sub-nanoscale location accuracy.

Reviewer #3

(Remarks to the Author)

The authors have revised their manuscript in response to my comments. Overall, I think the manuscript has been significantly improved. The added new table S2 in the supplementary materials is extremely helpful in highlighting the novelty of the work and how it compares to previous works (to me, the temporal resolution is the key aspect). I would suggest to move this table to the end of the main text, and adding the operating temperatures of each method as a separate column in the table.

I believe the manuscript now could be a good fit for publication in Nature Communications.

Response to the Reviewers

Reviewer #2 (Remarks to the Author):

It is good to see the authors refer to the recommended papers. However, spatially and temporally correlated sensing of local electric environment has been reported in many papers, besides Nat. Photon. 18, 230–235 (2024), see also Tom Delord, Richard Monge, and Carlos A. Meriles, Nano Letters 2024 24 (22), 6474–6479, which presents exact “simultaneous correlations in space and time” with NV system. There are neither relevant comments nor citations in the manuscript. Given the importance of the research area, there could be more pioneer works. I strongly suggest that the authors make systematic explorations in literature and comment on proper articles in the maintext. Again, the manuscript is not as claimed as “first time-resolved quantum electrometer working at the atomic scale”, neither the arXiv submission 2401.14290v1.

To address the reviewer’s concerns, we removed the ‘first-time’ adjective from the manuscript. We have also added the reference to the main text, which we were not aware of because our original submission was before publication. At the same time, we would like to iterate that the mentioned works do not provide subnanometer precision in the relative radial distance. And our original claim that reaching atomic scale/Angstrom scale resolution using linear Stark effect based sensors such as NVs will only be possible in extremely clean samples still remains.

Besides the insufficient novelty, the main issue lies in the non-solid conclusion and unconvincing analysis in the simulation built on unreasonable assumptions, for example, limiting the placement of the three proximity traps to a plane. Why do the traps/charges have to be in a plane? Is it a specially patterned implanting?

To address some of the reviewer’s concerns, the claims regarding the exact positioning and localization of the traps have been significantly toned down in the manuscript. We agree that confining the charges to a plane limits the generality of the simulations. However, we would like to point out that this is not a limitation of the method itself, as the confinement could have been relaxed at the expense of longer simulation times. Such a step, however, would not eliminate the uncertainty in the polar angle distribution, as shown in Fig. 2c of the manuscript.

We also want to highlight that only one of the peaks in the multimodal spectrum, the most red-shifted one, depends on all three absolute trap positions. Two of the peaks arise from interactions involving only two charged traps and the SnV, which are naturally confined to a plane. For those peaks, at least in terms of the local configuration of traps, no constraints need to be placed on the parameters to allow for meaningful relative positioning.

Below, we also remark that spherical symmetry with respect to the local probe is further broken by the remote traps and their distribution.

What is the precision in terms of scattering of the implantation in the x-y plane, is it better than the claimed location resolution? Same to the non-general choice of $r_2 = (0, 0, r_2)$.

The point the reviewer makes here is not entirely clear. The precision of the implantation is not related to the distribution of the traps that form after the sample has been annealed. During annealing, traps can diffuse. The localization will be entirely dependent on the relative positioning of those traps in relation to the SnV.

The assumptions are intensively made to simply the parameter space, but lead to unrealistic results, being lack of universality. Without the assumptions, there would be multiple solutions in the configuration of traps. Furthermore, how is the sign of r_2 determined?

We disagree with the claim that the assumptions lead to unrealistic results. In fact, the simulations can be viewed as a consistency check, demonstrating that the observed spectra are generally consistent with the assumption that charge traps near a defect, together with a distribution of remote traps, can produce the observed spectra. Configuration of the nearby traps are further verified by the temporal analysis of the acquired PLE data. We also object the reviewer's claim that the assumptions were made merely to simplify the parameter space. The assumptions are based on SRIM and annealing simulations of the damage caused and retained in the lattice after annealing. Not assuming a broken symmetry in the z-direction with respect to the annealing damage would have been unrealistic.

The sign of r_2 was determined by performing a Monte Carlo simulation with a coarse scan of the azimuthal angle θ_2 .

As also pointed out, the authors should address in detail how come the full location of a charge in 3-dimension space with a single-point detector. It is trivial, a full location needs a radial distance, the polar angle, and the azimuthal angle, or three coordinates. In the section of "Multimodal spectra" in Supplementary Material, however, only a single parameter (θ_i , $i=1,3$, claimed as azimuthal angle) representing angle is found. How is the other determined?

Here, we reiterate that we significantly toned down the claim of full localization in the manuscript. We still want to emphasize that symmetry is broken by the remote trap configurations, and that our simulations are based on this assumption. Breaking spherical symmetry in this way will also impact the possible trap configurations that we find in the simulations. Fixing the z-axis significantly reduces the number of dependent variables when simulating one or two traps. For example, confining the SnV and two other traps to a plane then represent all other possible polar angle configurations. As shown in Fig. 2c, the uncertainty in the polar angle was clearly visualized. During the simulation campaign, we fixed the azimuthal angle θ_2 using a coarse scan with the Monte Carlo method. We note that this choice is only relevant for fixing the orientation of the proximity traps with respect to the symmetry axis of the distribution that determines the placement of the remote traps, which in turn defines the direction of the z-axis. If the problem consisted of only three charges and no remote traps, the choice of the z-axis and its direction could always be made to coincide with the direction of r_2 . Setting up the problem this way makes θ_3 a dependent variable, as indicated in the supplementary material.

By the way, it is suggested that the authors mark out the angle in figures, e.g., fig. 2c, to avoid confusion.

Thank you for pointing that out. The angle has now been added to the mentioned plot and now detailed in the body of the manuscript.

The authors claimed that "no other effects contribute to the broadening" but failed to provide relevant experimental data. First of all, optical excitation power is a common factor inducing spectral broadening. It is necessary to provide the experimental test and make sure that there is no such broadening when determining the density of surrounding charges.

It is correct that the additional power broadening is not measured and calibrated in this work. Since the emitter's signal is by choice not stable due to the spectral jumps, it was not possible to directly conduct this measurement. The resonant saturation power, however, has been measured to be 4.2 μW in a tin-vacancy nanopillar sample with a similar preparation recipe (Trusheim et al. PRL 124 (2), p. 23602, 2020). Power broadening follows the algebraic equation $\Gamma_{\text{broadened}} =$

$\gamma_{\text{natural}} \times \sqrt{1 + \frac{\text{power}}{\text{power} + \text{power}_{\text{saturation}}}}$ (Cohen-Tannoudji et al., ISBN 978-0-471-29336-1, 1998). Measurements at saturation power experiences ~22% broadening. Since, our resonant power was selected 0.5 nW, and assuming an under estimated saturation power of 10 nW, the additional power broadening would be less than 2.5% and therefore negligible. To address the reviewer's concern, we added this analysis to the supplementary section 'Current limitations of the electrometer and how to overcome them'.

In terms of the strain dynamics, the author referred to the work Ji et al, Nat. Photon. 18, 230–235 (2024). However, this is in general related to the quality of crystal lattice and can be hugely different from sample to sample, especially between the ones dealt with laser-writing (used in the reference) and those of implantation (this work).

It is true that we have not explicitly measured a dynamical strain environment. Such an effect however has not been reported quantitatively, and it is not clear what dynamic strain range would even result in observable broadening effects. Since there are many reported measurements of life-time limited lines without any dynamical broadening in implanted samples, we can at least claim there were many cases in which such an effect did not play a role in any broadening effects. To address the reviewer's concern, we added the note that we have not tested for any dynamical strain environment and it can be an additional source of broadening the supplementary section 'Current limitations of the electrometer and how to overcome them'. Furthermore, For the GeV [Z. Li et al., Nat. Photon. 18(10), 1113–1120 (2024)], dynamic strain stemming from changes in the Jahn-Teller configuration of surrounding defects has been ruled out as the source of the observed spectra due to a significant mismatch between the predicted and observed hopping rates.

In summary, I cannot recommend the publication before the above questions are well addressed. Without determinative solutions, it is unpersuasive to make statements of “full positioning” and sub-nanoscale location accuracy.

We thank the reviewer one's more for the detailed comments, which we have addressed in the majority of the cases and which have improved the quality of our manuscript. To address the reviewer's concerns, we have reduced the claims regarding the sensor performance in the manuscript.

Reviewer #3 (Remarks to the Author):

The authors have revised their manuscript in response to my comments. Overall, I think the manuscript has been significantly improved. The added new table S2 in the supplementary materials is extremely helpful in highlighting the novelty of the work and how it compares to previous works (to me, the temporal resolution is the key aspect). I would suggest to move this table to the end of the main text, and adding the operating temperatures of each method as a separate column in the table.

We thank the reviewer for the positive evaluation and the additional suggestion. The table has now been moved to the main text, and the temperature information has been added.

I believe the manuscript now could be a good fit for publication in Nature Communications.

We thank the reviewer for their recommendation for publication.